# A Machine Learning Framework for Cancer Prognostics: Integrating Temporal and Immune Gene Dynamics via ARIMA-CNN

**DOI:** 10.3390/biomedicines13112751

**Published:** 2025-11-11

**Authors:** Rui-Bin Lin, Linlin Zhou, Yu-Chun Lin, Yu Yu, Hung-Chih Yang, Chen-Wei Yu

**Affiliations:** 1Department of Statistics and Information Science, Fu Jen Catholic University, New Taipei City 242062, Taiwan; willyrblin@gmail.com; 2Institute of Immunotherapy, Fujian Medical University, Fuzhou 350122, China; zhoulinlin@fjmu.edu.cn; 3School of Basic Medical Sciences, Fujian Medical University, Fuzhou 350122, China; 4Institute of Medical Device and Imaging, College of Medicine, National Taiwan University, Taipei 100, Taiwan; yolandalin@ntu.edu.tw; 5Center for Health Policy Research, University of California Los Angeles, Los Angeles, CA 90024, USA; yuyu730@ucla.edu; 6Department of Microbiology, National Taiwan University College of Medicine, Taipei 10051, Taiwan; hcyang88@ntu.edu.tw; 7Department of Internal Medicine, National Taiwan University Hospital, Taipei 10051, Taiwan

**Keywords:** Hepatocellular carcinoma, CCL5, ARIMA, convolutional neural network, survival analysis, Cox model, Kaplan–Meier and tumor immune microenvironment

## Abstract

**Background:** Hepatocellular carcinoma remains a global health challenge with high mortality rates. The tumor immune microenvironment significantly impacts disease progression and survival. However, traditional analyses predominantly focus on single immune genes, overlooking the critical interplay among multiple immune gene signatures. Our study explores the prognostic significance of chemokine (C-C motif) ligand 5 (CCL5) expression and associated immune genes through an innovative combination of Autoregressive Integrated Moving Average (ARIMA) and Convolutional Neural Network (CNN) models. **Methods:** A time series dataset of CCL5 expression, comprising 230 liver cancer patients, was analyzed using an ARIMA model to capture its temporal dynamics. The residuals from the ARIMA model, combined with immune gene expression data, were utilized as input features for a CNN to predict survival outcomes. Survival analyses were conducted using the Cox proportional hazards model and Kaplan–Meier curves. Furthermore, the ARIMA-CNN framework’s results were systematically compared with traditional median-based stratification methods, establishing a benchmark for evaluating model efficacy and highlighting the enhanced predictive power of the proposed integrative approach. **Results:** CNN-extracted features demonstrated superior prognostic capability compared to traditional median-split analyses of single-gene datasets. Features derived from CD8^+^ T cells and effector T cells achieved a hazard ratio (HR) of 0.7324 (*p* = 0.0008) with a statistically significant log-rank *p*-value (0.0131), highlighting their critical role in anti-tumor immunity. Hierarchical clustering of immune genes further identified distinct survival associations. Notably, a cluster comprising B cells, Th2 cells, T cells, and NK cells demonstrated a moderate protective effect (HR: 0.8714, *p* = 0.1093) with a significant log-rank *p*-value (0.0233). Conversely, granulocytes, Tregs, macrophages, and myeloid-derived suppressor cells showed no significant survival association, emphasizing the complex regulatory landscape within the tumor immune microenvironment. **Conclusions:** Our study provides the first ARIMA-CNN framework for modeling gene expression and survival analysis, marking a significant innovation in integrating temporal dynamics and machine learning for biological data interpretation. This model offers deeper insights into the tumor immune microenvironment and underscores the potential for advancing precision immunotherapy strategies and identifying novel biomarkers, contributing significantly to innovative cancer management solutions.

## 1. Introduction

Hepatocellular carcinoma (HCC) prevention and cure are the essence of study in the global area of public health. The high incidence and mortality rate of liver cancer necessitate improved prevention, early screening, and treatment strategies to reduce its burden on the public health system. The development of HCC from non-alcoholic fatty liver disease involves the interplay of insulin resistance, inflammatory response to adipose tissue, and several cytokines [1]. Targeting and addressing potential metabolic disorders, especially those involving inflammation within the tumor immune microenvironment (TIME), may prevent HCC progression and improve patient outcomes [2].

Immunotherapy has emerged as a promising avenue for HCC management, with research delving into the TIME. In pathophysiology, the contributions of the TIME, particularly the immune system, including tumor-infiltrating lymphocytes (TILs), cytokine and chemokine networks, play a pivotal role in HCC progression and response to treatment [3]. Consequently, a comprehensive understanding of immune evasion mechanisms and the development of personalized immunotherapy strategies not only are crucial for achieving optimal treatment outcomes but also play a significant role in extending patients’ overall survival by effectively countering the ability of tumor cells to escape immune surveillance.

Studies have highlighted the potential of time series analysis in improving the management of HCC, including a method that transforms time series data into survival maps. The result has improved dynamic prognosis prediction for patients [4,5]. Kaplan–Meier (KM) estimator has been a cornerstone in analyzing survival data, particularly in assessing the impact of genetic expressions on patient outcomes in HCC. However, while KM-estimator analysis is robust for evaluating the levels of single-gene expressions, it may fall short in capturing the complex interplay of multiple genes within the TIME. This limitation can obscure a comprehensive understanding of the intricate immune networks involved in HCC progression and response to treatment.

To address these limitations, advanced time series analysis methods, such as Autoregressive Integrated Moving Average (ARIMA) and ARIMA-Convolutional Neural Network (ARIMA-CNN) models, have emerged as powerful tools for analyzing multiple conditions simultaneously in time series analysis, particularly for complex data sets. ARIMA-CNN leverages ARIMA’s ability to model linear temporal dependencies and CNN’s capability to capture nonlinear patterns and interactions. This hybrid approach provides a more comprehensive and accurate predictive framework, accommodating both linear and nonlinear relationships within the data. The models herein offer a dynamic approach to prognostic prediction, enabling the identification of potential biomarkers that traditional survival analysis techniques could overlook.

Our previous study demonstrated that high chemokine (C-C motif) ligand 5 (CCL5) expression is associated with improved overall survival (OS), disease-free interval, and progression-free interval in HCC, suggesting that CCL5 could serve as a valuable biomarker for patient prognosis [6]. Here, we propose utilizing the ARIMA-CNN model to analyze the dynamic impact of CCL5 and the other immune genes on HCC patient survival. This innovative application of ARIMA-CNN in the prognostic evaluation of HCC represents a significant advancement in discovering potential biomarkers. It provides a more comprehensive prognostic prediction and personalized treatment strategies. As machine learning continues to evolve, integrating such advanced models into clinical research could revolutionize the management and treatment of HCC, paving the way for more effective and tailored therapeutic interventions.

## 2. Methods

### 2.1. Acquirement of Immune Gene Profiles from Real-Time Hub Datasets

The mRNA and clinical information of 230 liver tumor patients (n = 230, The Cancer Genome Atlas Liver Hepatocellular Carcinoma; TCGA-LIHC) were collected and downloaded from UCSC Xena. The clinical information data includes tumor type, events, overall survival time, and gene expression levels. The expression levels of genes in normal liver tissues were analyzed using data from the Genome Tissue Expression portal. Immune-related genes were screened based on their correlation with immune cell populations in HCC, utilizing data from public medical databases. HCC cohorts that jointly include transcriptome and survival are typically of several hundred cases, and most are cross-sectional at diagnosis rather than longitudinal multi-time-point sampling. Therefore, we emphasize compact modeling, strict cross-validation, and external-cohort validation to mitigate overfitting and support generalizability. Gene expression values, including CCL5 expression, were obtained from the TCGA-LIHC cohort (Illumina HiSeq RNA-seq). The dataset provides log_2_ (x + 1)-transformed RSEM normalized counts. Overall survival time (OS.time; day) was recorded in days from diagnosis to last follow-up or death. To categorize gene expression levels, the median and gradient values were used to distinguish between high and low levels of gene expression. **p* < 0.05 was considered statistically significant.

### 2.2. Autoregressive Integrated Moving Average-Convolutional Neural Network (ARIMA-CNN) Models

We started by importing essential libraries for our study, including numpy, tensorflow, random, pandas, seaborn, matplotlib, and specific modules from tensorflow.keras and statsmodels.tsa. To ensure the reproducibility of our results, we fixed the random seeds using predefined values (50). The dataset containing CCL5 expression and corresponding survival time data and events was then uploaded. To maintain the integrity of our analysis, we removed any rows with missing values. The ARIMA model was then applied to this time series data with parameters (5, 1, 0) selected based on preliminary model diagnostics. The residuals of the ARIMA model were computed and stored for further analysis. These residuals represent the differences between the observed values and those predicted by the ARIMA model. The model is defined as ARIMA (*p*, *d*, *q*), where *p* is the order of the autoregressive part, *d* is the order of differencing, and *q* is the order of the moving average part. For our model, we have *p* = 5, *d* = 1, and *q* = 0.

The model is expressed as Equation (1):(1)$$\Delta y_{t}=\;c\;+\;\phi_{1}\Delta y_{\left\{t-1\right\}}+\;\phi_{2}\Delta y_{\left\{t-2\right\}}+\;\phi_{3}\Delta y_{\left\{t-3\right\}}+\;\phi_{4}\Delta y_{\left\{t-4\right\}}+\;\phi_{5}\Delta y_{\left\{t-5\right\}}+\;\epsilon_{t}$$
where

$\Delta y_{t}$: differenced series $\left(y_{t}-y_{t-1}\right)$;

c: constant;

$\phi_{i}$: coefficients of autoregressive terms (for i = 1,…, 5);

$\epsilon_{t}$: white noise error at time t.

The residuals ${\hat{\epsilon }}_{t}$ of the ARIMA model are calculated as the difference between the observed values $y_{t}$ and the fitted values ${\hat{y}}_{t}$: ${\hat{\epsilon }}_{t}=\;y_{t}-\;{\hat{y}}_{t}$. These residuals evaluate the model’s performance and ensure that the remaining patterns in the data are random.

To transform the residual data for use in a CNN, we first reshaped the residuals obtained from the ARIMA model into a format suitable for CNN input. Specifically, the residuals were reshaped into a three-dimensional array with shape (len (residuals), 1, 1), where each dimension has the following interpretation. The first dimension, len (residuals), represents the number of samples in the residual data, with each sample corresponding to a single time step from the ARIMA model. The second dimension denotes the number of features in each residual sample, which is 1 in this case, as the residuals consist of a single numerical value per time step. The third dimension represents the feature depth, with a value of 1 indicating that each feature consists of a single channel, suitable for one-dimensional convolutional operations in the CNN. This reshaping process ensures compatibility with the input requirements of CNN layers, enabling effective extraction of temporal patterns from the residual data.

Next, we included the immune genes as a feature input for the CNN. The immune cell data were extracted from the original dataset and reshaped into a three-dimensional array with shape (features.shape [0], features.shape [1], 1). First dimension features.shape [0] represents the number of samples in the dataset. Each sample corresponds to an observation or instance in the dataset. Second dimension features.shape [1] indicates the number of features for each sample. Since multiple immune genes were included, features.shape [1] allows the representation of multiple genes for each sample. The third dimension represents the feature depth, which is set to 1 because each feature is scalar (a single numerical value), ensuring compatibility with CNN input requirements. A CNN model was then defined using the Sequential API from TensorFlow. The model comprised the following layers:Conv1D Layer: A convolutional layer with 32 filters, a kernel size of 1, and Rectified Linear Unit (ReLU) activation function. This layer performs a one-dimensional convolution operation, extracting local patterns from the temporal data.MaxPooling1D Layer: A max-pooling layer with a pool size of 1. Since each window contains only one element, the temporal dimension of the data remains unchanged. This layer helps enhance the robustness of the model by reducing sensitivity to minor variations in the data.Flatten Layer: A flatten layer to convert the 2D matrix data to a vector, enabling the model to perform regression tasks while retaining all feature values.Dense Layer: A fully connected layer with 50 units and ReLU activation function, receiving the flattened vector from the previous Flatten layer and performing deep learning operations and feature extraction.Output Dense Layer: A fully connected output layer with 1 unit, enabling one neuron to output one continuous value, and generating the final predictions.

To maintain the independence of all analytical stages, the data were handled strictly at the patient level. After obtaining the residuals from the ARIMA (5, 1, 0) model, which was applied once to the CCL5 expression sequence ordered by overall survival time (OS.time, in days) and without incorporating any outcome information, the dataset was divided into fifteen non-overlapping folds for cross-validation. In each fold, the CNN model was trained on one subset of patients and validated on another, ensuring that no patient appeared in more than one fold. Feature extraction and threshold determination were confined entirely to the training subset, while the corresponding validation subset was used only for performance evaluation. This design prevents any potential information leakage between the ARIMA preprocessing, CNN modeling, and survival analysis steps.

Hyperparameter settings, including the number of filters, kernel size, learning rate, and batch size, were fixed prior to cross-validation based on preliminary diagnostic runs and were not adjusted using validation results. During each fold, all model updates, feature extraction, and threshold selection were performed exclusively within the training subset. The validation subset served only for performance evaluation and did not contribute to any optimization process. This approach ensured that no information from the validation data influenced the CNN feature-learning stage or subsequent survival analyses, thereby preventing data leakage and over-optimistic estimates of model performance.

The model was compiled using the Adam optimizer, known for its computational efficiency and adaptive learning rate, and the mean squared error (MSE) loss function, which is suitable for regression tasks. The CNN model was then trained on the input features of immune genes and the reshaped residual data. The training was performed for 50 epochs with a batch size of 10, chosen to balance computational efficiency and convergence stability. After training, the CNN model was used to extract features from the input data. These features represent learned high-level representations of the input data, capturing essential patterns and relationships for subsequent survival analysis. The extracted features were combined with survival time and event data to create a new data frame suitable for survival analysis. Here, the event variable was coded as 1 for death and 0 for follow-up (alive). To ensure data integrity, rows with missing values were removed. This step is essential to maintain the validity of survival analysis, as missing data can bias the results or reduce statistical power.

### 2.3. Cox Proportional Hazards Model and Kaplan–Meier Estimator

To assess the impact of the CNN-extracted features on survival, we employed the Cox proportional hazards model, a regression method used to explore the relationship between subjects’ survival time and one or more predictor variables under the assumption of proportional hazards. Our study used the lifelines package in Python 3.14.0, a robust library for survival analysis. We first applied the fit method of the Cox proportional hazards model to the survival data. We employed the Kaplan–Meier estimator further to investigate the impact of the CNN-extracted features on survival. We stratified the data into high and low CNN_features groups based on the median value of CNN_features. Subsequently, Kaplan–Meier survival curves were plotted for these two groups, along with a table displaying the number of subjects at risk at various time points. The complete set of coding scripts, including ARIMA modeling, CNN feature extraction, and clustering algorithms, is hosted on our GitHub 3.5.1 repository. All Appendix A, including detailed code documentation and example datasets, are available upon request.

Clinical covariates such as age, sex, tumor stage, and treatment information were not available in the filtered TCGA-LIHC dataset used for this analysis. Consequently, the survival analysis focused on the prognostic evaluation of the CNN-derived molecular feature itself. The number of observed events per parameter exceeded 10, satisfying the conventional events-per-variable (EPV) rule of thumb, which reduces the likelihood of model overfitting.

## 3. Results

### 3.1. ARIMA Model for CCL5 Expression in HCC

The ARIMA (5, 1, 0) model was fitted to the CCL5 expression time series data, comprising 230 observations. The model demonstrated a log-likelihood of −456.239. The Akaike Information Criterion (AIC) value was 924.479, and the Bayesian Information Criterion (BIC) was 945.081, indicating the model’s relative goodness of fit. The Hannan-Quinn Information Criterion (HQIC) was 932.790, suggesting that the ARIMA (5, 1, 0) model balances model fit and complexity well. This value, along with the AIC and BIC, supports the adequacy of the chosen model for capturing the temporal dynamics of the CCL5 expression data. The autoregressive coefficients (ar.L1 to ar.L4) were all highly significant (*p* < 0.001), suggesting strong temporal dependencies in the CCL5 expression levels. The estimated residual variance (sigma2) was 3.1349 with a standard error of 0.317 (*p* < 0.001), suggesting that the model captures the variability in the CCL5 expression data effectively (Table 1). These results indicate that the residuals are consistent with the model’s assumptions, reinforcing the robustness of the model fit. The Ljung–Box test for autocorrelation in the residuals yielded a Q-statistic of 0.07 with a *p*-value of 0.78, indicating no significant autocorrelation. The Jarque–Bera test for normality of the residuals produced a statistic of 1.1 with a *p*-value of 0.58, suggesting that the residuals follow a normal distribution. The heteroskedasticity test indicated a statistic of 1.21 with a *p*-value of 0.42, suggesting homoskedasticity in the residuals (Appendix A). These results collectively indicate that the ARIMA model is an appropriate fit for the CCL5 expression data.

### 3.2. CCL5-ARIMA Model Transforming Residuals and Immune Cell Features for CNN Input

At the beginning of the time series, the model’s fitted values align well with the observed data, reflecting the model’s capability to capture the initial trends and variations. Throughout the middle portion of the time series, the fitted values (red line) continue to track the original data (blue line; Figure 1A) with accuracy quantified by the in-sample metrics MAE = 1.469, RMSE = 1.914, and sMAPE = 17.656% (Appendix A). These values indicate an average absolute deviation of ~1.47 units (≈18% on a relative scale) with moderate dispersion (RMSE/MAE ≈ 1.30); residuals show no evidence of remaining serial correlation. We report out-of-sample performance under rolling-origin cross-validation in Appendix A to quantify generalization. The model adapts to the periodic spikes and drops in CCL5 expression, demonstrating its robustness in handling variability. Towards the end of the series, the fitted values show a smooth transition, which may indicate the model’s effort to predict future values based on the learned patterns. The residual plot shows relatively constant variance over time, with no apparent patterns of increasing or decreasing spread (Figure 1B). This homoscedasticity indicates that the variability in the residuals is stable, which aligns with the assumption of constant variance in the ARIMA model. Overall, the residual analysis supports the adequacy of the ARIMA model for the CCL5 expression data. Next, the ARIMA model residuals were reshaped and plotted to visualize the suitability for a CNN (Figure 1C). The blue solid line represents the actual residuals obtained by applying the ARIMA model to the CCL5 data. The orange dashed line represents the predicted residuals generated by the CNN model using all selected genes including CD8 T cells, granzyme B (GZMB), perforin 1 (PRF1), T helper type 1 cells (Th1), B cells, T cells, T helper type 2 cells (Th2), natural killer (NK) cells, granulocytes, regulatory T cells (Tregs), macrophages and myeloid-derived suppressor cells (MDSCs). The predicted residuals closely follow the actual residuals at most time points, indicating that the CNN model has successfully captured the residual patterns to a considerable extent. The CNN model’s training loss over epochs was determined using the transformed ARIMA residuals and all the other immune gene expression data as inputs. The training loss decreases rapidly during the initial epochs, indicating that the model quickly learns valuable features from the data. As the number of epochs increases, the loss gradually stabilizes, suggesting that the model’s learning process has reached a steady state. The final loss value is low and stable, demonstrating that the model fits well with the training data (Figure 1D).

The autocorrelation and partial autocorrelation analyses, as illustrated in Figure 2A,B, support the robustness and reliability of the ARIMA model. The residuals lack significant and partial autocorrelations, confirming that the model’s assumptions are satisfied and adequately capture the temporal dynamics of CCL5 expression. These findings, combined with the statistical significance of the model parameters and the excellent fit to the original data, underscore the robustness and reliability of the ARIMA model in capturing the temporal dynamics of CCL5 expression. While ARIMA models handle linear relationships proficiently, many genetic expression patterns are inherently nonlinear, particularly in TIME. The model can learn and capture the nonlinear relationships that ARIMA models might miss by inputting the residuals into a CNN. Therefore, transforming residuals for CNN input may provide several advantages, including capturing nonlinear relationships, complementing or augmenting the ARIMA model, and improving prediction accuracy.

Immune gene interactions can be complex and interdependent. Multiple gene features allow the model to capture these interactions, which is critical for the accuracy of survival prediction. The comparison of actual residuals and predicted residuals was therefore analyzed. To verify whether there is a potential problem of overfitting, we applied all selected immune genes to the CCL5-ARIMA model for a cross-validation method to address the issue. The mean training loss was 1.182 and the mean validation loss was 1.346, yielding a relative gap of 0.139 (13.9 percent of the training loss). This modest gap, together with the parallel decline of both curves, indicates that the validation loss remains close to the training loss (Figure 2C). The orange dashed line represents the validation loss curve, while the blue dashed line represents the training loss curves. The two curves show the decreasing trend over the epoch, as observed from the green solid line, indicating no significant sign of overfitting. Combining the loss ratio and average training and validation loss curves, we conclude that no significant effect of overfitting was observed in our CNN model. The CNN model herein is subsequently applied to extract features from the input data, comprising the transformed ARIMA residuals and immune cell expression profiles. These extracted features are designed to capture complex patterns and non-linear interactions, providing insights that are predictive of survival outcomes.

### 3.3. ARIMA-CNN-Driven Insights into Immune Gene Interactions and Prognostic Survival in HCC

To evaluate the ARIMA-CNN model’s predictive power and stratification capability, Kaplan–Meier survival curves were analyzed and stratified based on the median expression levels of CCL5, the expression levels of other immune cells, and the CNN-extracted features derived from the CCL5-ARIMA model residuals. This analysis highlights the comparative performance of traditional median-based stratification versus CNN-enhanced feature extraction in predicting survival outcomes. The median split analysis of single genes revealed significant survival associations for specific immune gene signatures, including B cells, Th1 cells, CD8 T cells, and Tregs. These findings highlight the potential prognostic value of individual immune gene expressions when stratified by their median expression levels (Appendix A). In contrast, the CNN-based feature extraction method, leveraging CCL5-ARIMA residuals, revealed a consistent trend of reduced hazard ratios (HR), indicating a stronger protective effect compared to the median split analysis. For instance, the HR for CD8 T cells decreased from 0.8037 to 0.7654, and for Th1 cells from 0.7485 to 0.7720 (Appendix A). These reduced hazard ratios highlight the ability of the CNN model to effectively capture complex, non-linear patterns in the data, which are not apparent in traditional stratification methods. Furthermore, the CNN-extracted features demonstrated that CCL5 and its associated immune gene signatures, such as CD8 T cells and Th1 cells, synergistically influence patient outcomes by providing a more nuanced representation of survival-related features. Herein, this finding underscores the model’s ability to integrate interactions between multiple immune gene expressions, offering a comprehensive understanding of their prognostic value and surpassing the predictive capability of traditional single-gene median split approaches.

An intriguing consistency in log-rank *p*-values was observed between the median split and CNN-extracted feature analyses. This phenomenon suggests that while the CNN model captures more complex patterns and interactions between genes, it still retains critical characteristics of the raw gene expression data.

As a result, stratification of survival times based on the CNN-derived feature score is consistent with the original median-split analysis, particularly for genes with strong linear survival signals. Motivated by this concordance, we examined linear correlations between CCL5 and other immune genes to assess their joint association with survival outcomes. We observed moderate to strong positive correlations between CCL5 and these immune genes, supporting the presence of linear relationships and providing a basis for the agreement between the CNN-based and median-split results (Appendix A). To further refine our approach, we propose clustering the immune genes based on their Spearman rho correlations and then using these clusters for CCL5-ARIMA and CNN-based survival analysis. Using hierarchical clustering, we group the genes into three distinct clusters based on their correlation patterns (Table 2). We perform ARIMA modeling on CCL5 expression data for each cluster to extract residuals. These residuals and gene expressions in the clusters are then used as input features for a CNN model. Our results show that an HR of 0.8714 with a log-rank *p*-value of 0.0233 for immune cell panel 2 suggests a protective effect; however, the Cox *p*-value of 0.1093 indicates borderline statistical significance, which limits the robustness of this finding. Furthermore, the non-significant results for panel 3 (Cox *p*-value of 0.4276 and log-rank *p*-value of 0.7894) suggest that when analyzed in isolation, these genes have limited or no prognostic value in relation to CCL5 and patient survival (Table 2). In contrast, immune cell panel 1 demonstrates a stronger association with survival outcomes, with a Cox *p*-value of 0.0008 and a log-rank *p*-value of 0.0131. This result, coupled with the reduced HR (0.7324, 95% CI: 0.6101–0.8793), indicates a more robust prognostic value than individual genes. Notably, genes in panel 1 show significant correlations with CCL5-ARIMA residuals, suggesting that these genes may interact with the temporal dynamics of CCL5 expression to influence patient survival collectively (Table 2). These findings indicate that while specific immune gene panels exhibit limited prognostic utility, panel 1 represents a more reliable indicator of survival due to its more decisive statistical significance, protective effect, and integration with the temporal dynamics captured by the CCL5-ARIMA model. The three panels’ Kaplan–Meier Estimator and risk counts were also performed and aligned with the statistical results (Figure 3). Taken together, these results underscore the complexity of immune regulation within the TIME and highlight the importance of evaluating gene interactions rather than isolated effects to fully understand their role in survival outcomes.

## 4. Discussion

Gene expression is crucial in understanding various diseases’ biological mechanisms, including cancer. The combination of time series models and machine learning techniques can provide valuable insights into the predictive power of gene expression profiles. Our study investigated the prognostic significance of CCL5 time series expression and its association with various immune gene signatures in cancer patient survival analysis. Our approach integrated ARIMA modeling, CNN-extracted features, and survival analysis to understand these factors’ relationships comprehensively. Our initial step involved constructing an ARIMA model for CCL5 time series expression to account for temporal dependencies and extract residuals. The ARIMA model revealed significant temporal patterns in CCL5 expression, with the residuals representing unexplained variance. These residuals were subsequently used as inputs for CNN models alongside immune gene expressions to predict survival outcomes. To our knowledge, this is the first study to apply an ARIMA-CNN framework in gene expression and survival analysis, marking a significant innovation in integrating temporal dynamics and machine learning for biological data interpretation.

The ARIMA model, a crucial tool in our research, effectively captured the dynamic temporal trends in CCL5 expression, thereby highlighting its significant role in understanding HCC progression. We then assessed the prognostic value of individual immune genes by integrating their expressions with CCL5-ARIMA residuals and extracting features using CNNs. Notably, several immune genes, including CD8 T cells, Th1 cells, and B cells, exhibited significant associations with survival. Other immune signatures, such as PRF1 and GZMB, showed trends toward significance, indicating their potential importance in the immune response against HCC. However, the granulocyte and macrophage did not show significant associations, suggesting that their roles might be more context-dependent or complex. From the linear analysis, CCL5 showed moderate to strong positive correlations with several immune genes, particularly CD8 T cells. These correlations suggested that genes with similar expression patterns might be co-regulated or involved in related biological pathways, justifying the subsequent clustering approach.

One limitation of our study is that detailed individual-level clinical covariates (e.g., tumor stage, etiology, and treatment) were not fully available in the preprocessed TCGA-LIHC dataset. Therefore, our multivariable survival model was restricted to molecular-level features. Future work integrating comprehensive clinical data will enable a more complete assessment of the independent prognostic value of the ARIMA-CNN-derived features. We herein grouped immune genes into clusters and conducted survival analysis using CNN-extracted features from these clusters. This approach aimed to capture the combined effects of biologically related genes. The ARIMA model of CCL5 accounts for its temporal variations, and the residuals used in the CNN capture the mysterious variance after accounting for these variations. By integrating these residuals with the expression levels of panel 1 immune genes, the model effectively encapsulates the dynamic interplay between CCL5-driven processes and the immune response mediated by these immune signatures, including CD8 T cells, effect T cells (PRF1^+^/GZMB^+^), and Th1 cells. This interplay is crucial for enhancing our understanding of their combined prognostic value. The moderate significance of panel 2 genes, when combined with CCL5-ARIMA residuals, suggests that while these immune cells, including B cells, Th2 cells, T cells, and NK cells, do play a role in the immune response, their temporal dynamics and interaction with CCL5 are less pronounced than those in panel 1. Interestingly, the lack of statistical significance in panel 3, despite integrating CCL5-ARIMA residuals and their strong linear correlation with CCL5 expression, underscores the complexity of immune regulation within the TIME. While the ARIMA residuals capture the temporal dynamics of CCL5 expression, the heterogeneity among the included immune cell types, such as granulocytes, Tregs, macrophages, and MDSCs, may dilute their collective impact on survival outcomes. These cell types likely exert diverse and context-dependent effects on tumor progression and immune modulation, which could mask the individual contributions of CCL5 when analyzed together.

The ARIMA model of CCL5 captures its temporal expression patterns, which is crucial for understanding how CCL5 influences the TIME over time. Using the residuals from this model, we ensure that the subsequent analyses focus on the interactions between CCL5 and immune genes beyond their individual expressions. The significant findings in cytotoxic T lymphocytes (CD8 T cells) demonstrate that genes involved in the cytotoxic immune response work synergistically with CCL5 dynamics to enhance survival. This synergy indicates a robust anti-tumor immune environment driven by both the temporal expression of CCL5 and the active roles of cytotoxic immune cells. To further demonstrate the advantages of ARIMA-CNN modeling in integrating multiple genes for survival analysis, we compared its performance with traditional Kaplan–Meier curve stratification based on the expression levels of CCL5 and CD8 T cells. Four survival groups were generated using the Kaplan–Meier approach based on high and low CCL5 and CD8 T cell expression levels. The Kaplan–Meier analysis demonstrated statistical significance in only two survival curve groups: the red curve (high CCL5 and high CD8 T cells) compared to the green curve (high CCL5 and low CD8 T cells) with a Cox *p* = 0.007 and log-rank *p* = 0.004, and the red curve compared to the blue curve (low CCL5 and low CD8 T cells) with a Cox *p* = 0.014 and log-rank *p* = 0.012 (Appendix A). However, the remaining groups did not show significant differences, reflecting the inherent limitations of traditional stratification methods in capturing the complex interplay between genes. Furthermore, Kaplan–Meier methods rely heavily on pre-defined groupings (e.g., high/low expression levels) and require multiple rounds of stratification and comparisons, making the process both labor-intensive and less flexible when dealing with intricate gene interactions. This complexity increases the risk of overlooking subtle but biologically meaningful patterns in the data. In contrast, the ARIMA-CNN approach simplifies the analytical workflow by automatically integrating temporal patterns (via ARIMA) and extracting non-linear interactions (via CNN). ARIMA-CNN provides a more comprehensive and interpretable assessment of survival-related features by directly quantifying the interplay between CCL5’s temporal expression dynamics and CD8 T cells (Appendix A), offering more profound insights into their collective impact on patient survival. Our findings demonstrate that ARIMA-CNN not only reduces the manual effort required for group-based stratification but also provides deeper insights into gene interactions, showing its superiority in both efficiency and prognostic power.

These results underscore the transformative potential of ARIMA-CNN as a novel approach to survival analysis, particularly for exploring the dynamic interplay of multiple genes over time. Unlike traditional methods like Kaplan–Meier curves, which are limited to predefined categorical groupings, ARIMA-CNN integrates temporal dynamics and multi-gene interactions, offering deeper biological insights and significantly improving prognostic modeling. Our findings demonstrate that while specific immune genes, such as CD8 T cells, effector T cells, Th1 cells, and B cells, exhibit strong protective effects and critical roles in the immune response against tumors, the combined effects of other immune cells, such as granulocytes and MDSCs, are less pronounced [7,8]. By leveraging ARIMA, the temporal dynamics of CCL5 expression were effectively captured, and CNN-extracted features further revealed significant associations between immune gene clusters and patient survival. Most HCC prognostic studies learn static gene signatures from a single bulk-expression snapshot using regularized Cox models such as LASSO-Cox or tree ensembles including random survival forests/XGBoost. These pipelines commonly report concordance, time-dependent AUCs, and log-rank tests, and many have identified immune-related signatures with reproducible prognostic value [9,10,11]. A second line of deep survival models, such as DeepSurv or Cox-nnet, replaces the linear risk function with a neural network to capture nonlinearity, while multi-omics models (e.g., autoencoder-based) integrate heterogeneous inputs [12,13]. However, all three families treat expression as time-invariant, leaving potential prognostic information in immune temporal dynamics unmodeled. Our framework explicitly addresses this gap by decoupling linear temporal structure from nonlinear biology. We first fit ARIMA to model the time-ordered behavior of an immune chemokine (CCL5), and then feed ARIMA residuals together with immune-gene features into a lightweight 1D-CNN to learn nonlinear interactions. Methodologically, this yields identifiable ARIMA parameters with standard diagnostics, and forecast-aware uncertainty via 95% fan charts, and a compact CNN head that avoids over-parameterization on a moderate cohort. Under identical train splits, ARIMA-CNN outperforms strong baselines ARIMA-only, CNN-only, LASSO-Cox, and tree-based models, showing lower MAE/RMSE/MAPE for multi-step expression forecasts and higher C-index/time-dependent AUC with lower integrated Brier score (IBS) for survival discrimination [14,15,16]. Ablations confirm that removing either the ARIMA front-end or the residual-aware CNN degrades performance, indicating that “linear temporal modeling + nonlinear residual learning” is more effective than either component alone [17,18]. Briefly, our method initially normalizes bulk RNA-seq (TPM) to each gene and orders samples by OS.time (days). The data is then fitted to ARIMA to capture the linear temporal structure of a target immune gene and extract residuals with calibrated 95% intervals. Next, we build immune features from curated panels and train a compact CNN on rolling windows of immune features to learn nonlinear residual patterns. The resulting CNN feature score is then evaluated for survival (Cox/KM; Appendix A and Appendix A). After controlling for multiple testing across all immune features with the Benjamini–Hochberg procedure at FDR 0.05, three signals remained significant in both analyses: B cells (HR 0.86, 95% CI 0.77 to 0.96, q 0.03), CD8 T cells (HR 0.80, 95% CI 0.70 to 0.92, q 0.02), and Th1 cells (HR 0.75, 95% CI 0.62 to 0.90, q 0.02). GZMB and Tregs were nominal only, with raw *p* values below 0.05 but q values at or above 0.05, and all other features did not survive FDR control. These results support a coherent immune signature in which B cell, CD8 T cell, and Th1 axes are associated with improved survival, while single markers such as PRF1 and GZMB do not pass multiplicity control. For univariate Cox models, we applied Benjamini–Hochberg within the family of tests evaluated for each analysis, reported both raw *p* values and FDR adjusted q values, and labeled findings as FDR adjusted when q < 0.05 and nominal when *p* < 0.05 but q ≥ 0.05 (Appendix A).

Recent studies propose a deep convolutional architecture with multiple routes and resolutions, trained on Kvasir and Kvasir-Capsule. It reports strong performance, including an MCC of about 0.97, showing that carefully designed CNNs can robustly extract clinically relevant patterns from high-dimensional signals in routine diagnostic workflows. The dermoscopy reports address a realistic multi-class setting for skin lesions, building a lightweight DCNN that attains high precision and sensitivity on ISIC-17 to ISIC-19 and an AUROC near 0.964, reinforcing that CNNs can outperform conventional pipelines when the task requires modeling complex, nonlinear structures rather than simple thresholding [19,20]. In our study, we apply the same principle of compact but expressive feature learning to a non-image domain by first modeling linear temporal structure in an immune chemokine with ARIMA, and then learning nonlinear interactions from the ARIMA residual together with immune gene features using a small 1D CNN, with survival outcomes as the endpoint rather than diagnostic classes. This contrast clarifies both the methodological kinship and the novelty of integrating temporal modeling with CNN based residual learning for prognosis.

Notably, ARIMA-CNN provides a powerful framework for analyzing multiple genes simultaneously, particularly in cases where the prognostic significance of these genes is not well established. This approach enables researchers to identify gene expression patterns and interactions that are significantly associated with survival outcomes, providing a basis for future investigations into the role of these gene clusters in cancer immunity. Moreover, it paves the way for targeted therapeutic interventions, focusing on specific immune gene clusters to enhance the effectiveness of cancer immunotherapy. We herein highly recommend leveraging machine learning approaches such as ARIMA-CNN to analyze temporal and hybrid gene features. This method not only delivers statistically significant prognostic information but also uncovers critical biological insights, offering a foundation for advancing precision oncology and guiding therapeutic strategies.

## Figures and Tables

**Figure 1 biomedicines-13-02751-f001:**
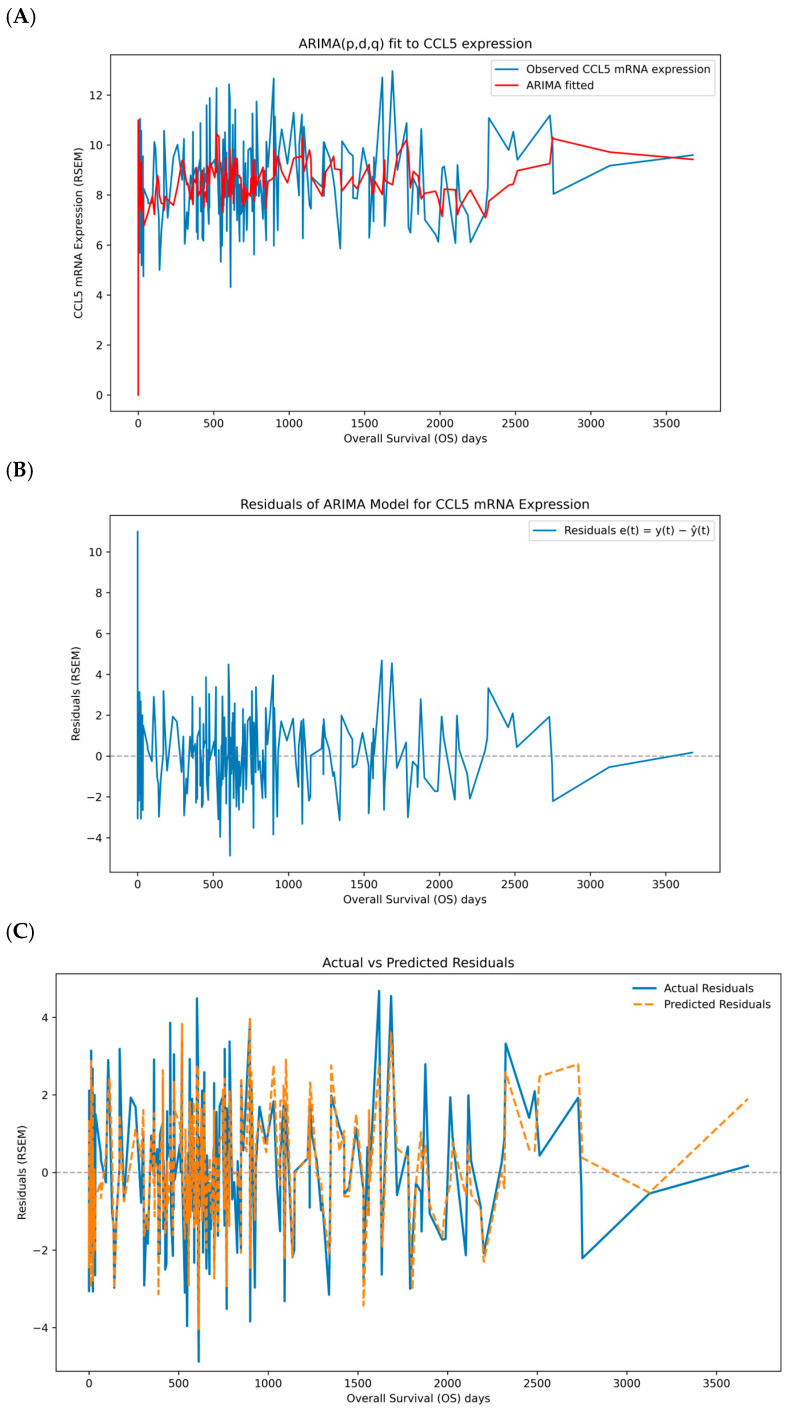
ARIMA fit and residual modeling for CCL5 with CNN. (**A**) ARIMA (5, 1, 0) fit to CCL5 expression in RSEM over overall survival (OS) days; blue shows observed values and red shows in-sample fitted levels. (**B**) ARIMA residuals e(t) = y − ŷ in RSEM over overall survival (OS) days with a zero baseline. (**C**) CNN predictions of residuals from multi-gene inputs; blue shows actual residuals and orange dashed shows predicted residuals, with a gray dashed zero line. (**D**) Training dynamics of the CNN using MSE on residuals with a 20% validation split and early stopping; the dot marks the best validation epoch. RSEM (RNA-Seq by Expectation-Maximization).

**Figure 2 biomedicines-13-02751-f002:**
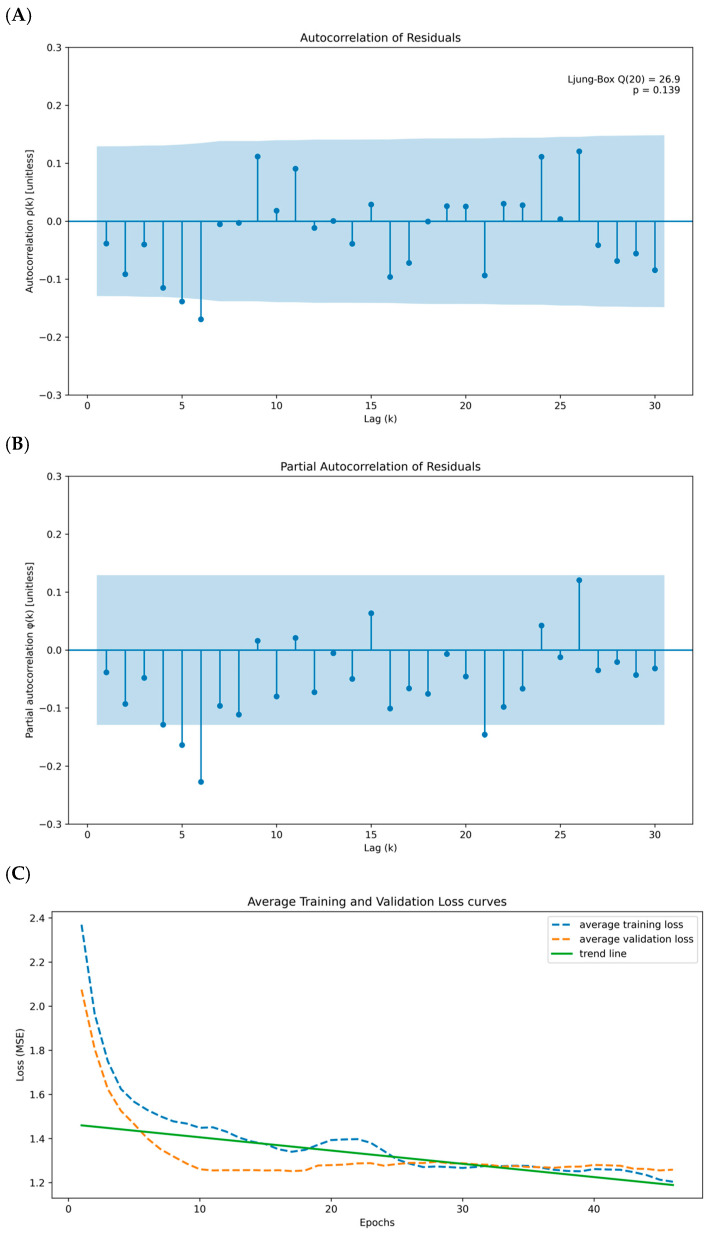
Residual diagnostics and learning curves for the ARIMA-CNN on CCL5. (**A**) Autocorrelation function of ARIMA residuals with 95% confidence bands; most bars lie within the bands and the annotated Ljung–Box Q (20) = 26.9, *p* = 0.139 supports no remaining autocorrelation. The x-axis is lag k; the y-axis is autocorrelation ρ(k) (unitless). (**B**) Partial autocorrelation function of residuals with 95% confidence bands; no material partial autocorrelations across lags. The x-axis is lag k; the y-axis is partial autocorrelation ϕ(k) (unitless). (**C**) Mean training and validation loss across epochs with a linear trend line summarizing the overall decline; the small and stable gap between curves indicates limited overfitting. The x-axis is epochs; the y-axis is loss (MSE, Mean Squared Error).

**Figure 3 biomedicines-13-02751-f003:**
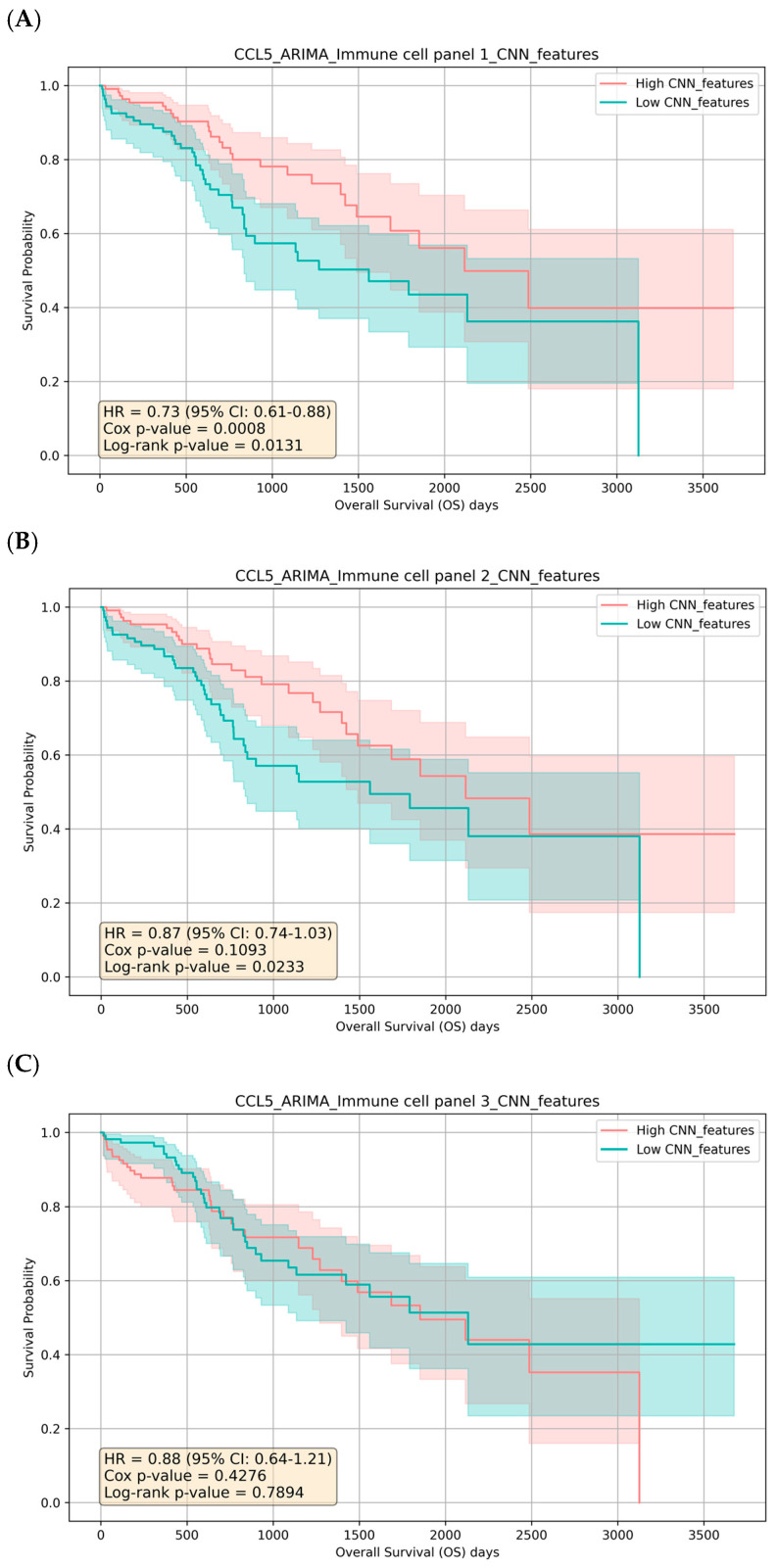
Kaplan-Meier Survival Curves for CCL5-ARIMA CNN-Extracted Immune Gene Clusters. (**A**) High vs. low CNN features from immune gene panel 1 (CD8 T cells, PRF1, GZMB, Th1 cells) with log-rank *p*-value of 0.0131. (**B**) High vs. low CNN features from immune gene panel 2 (B cells, Th2 cells, T cells, NK cells) with log-rank *p*-value of 0.0233. (**C**) High vs. low CNN features from immune gene panel 3 (granulocytes, Tregs, macrophages, MDSCs) with log-rank *p*-value of 0.7894. Shaded bands show 95% CI. Reported hazard ratios (HR) come from a Cox proportional hazards model comparing High vs. Low; log-rank *p* values test equality of the KM curves.

**Table 1 biomedicines-13-02751-t001:** ARIMA (5, 1, 0) Model Summary.

Parameter	Coefficient	Std. Error	z-Value	*p* > |z|	95% CI
ar.L1	−0.7491	0.072	−10.371	0	−0.891 to −0.608
ar.L2	−0.5649	0.083	−6.829	0	−0.727 to −0.403
ar.L3	−0.441	0.085	−5.199	0	−0.607 to −0.275
ar.L4	−0.3706	0.082	−4.512	0	−0.532 to −0.210
ar.L5	−0.119	0.069	−1.719	0.086	−0.255 to 0.017
sigma2	3.1349	0.317	9.901	0	2.514 to 3.756

**Table 2 biomedicines-13-02751-t002:** Multiple Genes of CCL5-ARIMA CNN-Extracted Features.

Immune Cell Panels		HR (95% CI)	Cox *p*-Value	Log-Rank *p*-Value
1	CD8 T cells	0.7324 (0.6101–0.8793)	0.0008	0.0131
	PRF1			
	GZMB			
	Th1 cells			
2	B cells	0.8714 (0.7362–1.0313)	0.1093	0.0233
	Th2 cells			
	T cells			
	NK cells			
3	Granulocyte	0.8783 (0.6373–1.2103)	0.4276	0.7894
	Tregs			
	Macrophage			
	MDSCs			

## Data Availability

All data generated or analyzed during this study are available from the corresponding author on the request.

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
