# Peer review of "A Machine Learning Framework for Cancer Prognostics: Integrating Temporal and Immune Gene Dynamics via ARIMA-CNN"

_biomedicines, 2025, doi:10.3390/biomedicines13112751_

Round 1
Reviewer 1 Report (Previous Reviewer 1)
Comments and Suggestions for Authors
There are are some missing-unclear parts in the paper. To give clear review they should be corrected and detailed.
Firstly,
The paper has lack of readability and clearness.
If it can corrected and detailed anaysis can be done about the figures.
otherwise it should be rejected.
"Figure 1. ARIMA model fit for CCL5 expression and the immune cell expressions for CNN." cannot be readable.
What do axis and ordinates mean?
Similarly
"Figure 2. Residual Analysis and Training Loss Evaluation for ARIMA-CNN on CCL5 Expression Data." in not clear.
At the same time
"Figure 3. Kaplan-Meier Survival Curves for CCL5-ARIMA CNN-Extracted Immune Gene Clusters." cannot be readable.
therefore they cannot be interpreted.
Secondly,
The paper is titled as "A Machine Learning Framework for ***". Therefore I want to see the framework of the model in the paper. In this framework, the authors should show integrated methodological and computational architecture that systematically combines multiple analytical components—such as data preprocessing, temporal modeling, feature extraction, and predictive learning—to achieve a unified goal of cancer outcome prediction.
Thirdly,
What are the discriminative features of the proposed work?
Although the authors said that "Our result is the first study to apply an ARIMA- CNN framework in gene expression and survival analysis, marking a significant innovation in integrating temporal dynamics and machine learning for biological data interpretation."
They should compare the results with other works or ML algorithms.
Fourthly,
The authors said that "the dataset was divided into fifteen non-overlapping folds for cross-validation"
Why do you prefer to divide your dataset to 15 parts. In Data science gereraly 5 fold, 10 fold approaches are used!
Explain your rationale.
Finally,
What do you mean with "based on CNN features aligns" CNN Features here?
Author Response
"Please see the attachment."

Reviewer 2 Report (Previous Reviewer 2)
Comments and Suggestions for Authors
Dear Editor,
Thank you for the opportunity to review this manuscript. I have carefully examined the revised version along with the authors' responses to the previous comments.
I am pleased to recommend this manuscript for publication in your journal, as the authors have effectively addressed my suggestions, resulting in significant improvements to the clarity, methodology, and overall scientific rigor of the study.
Furthermore, this research holds substantial significance within the scientific community, particularly in the domain of autoregressive integrated moving average. The proposed approach presents methodological advancements that can contribute to ongoing research and practical applications, making it a valuable addition to the literature.
I appreciate the opportunity to contribute to the peer review process and look forward to seeing this work published.
Thank you
Author Response
Thank you very much for the opportunity to take the time to review our manuscript. We are grateful for your kind support and acceptance.
Round 2
Reviewer 1 Report (Previous Reviewer 1)
Comments and Suggestions for Authors
The authors did the related corrections.
It can be accepted as is.
This manuscript is a resubmission of an earlier submission. The following is a list of the peer review reports and author responses from that submission.
Round 1
Reviewer 1 Report
Comments and Suggestions for Authors
The paper is titled as “A Machine Learning Framework for Cancer Prognostics: Integrating Temporal and Immune Gene Dynamics via ARIMA-CNN” and it is aimed to explore the prognostic value of immune-related genes in hepatocellular carcinoma by introducing a novel hybrid ARIMA-CNN framework.
The authors combine time-series modeling of chemokine (CCL5) expression with convolutional neural networks to predict survival outcomes and benchmark their approach against traditional median-based stratification methods.
The topic is interesting and it addresses the intersection of cancer immunology, time-series data analysis, and machine learning concepts. And it fits well within the scope of Biomedicines journal.
However, I have a number of concerns regarding rejection of the paper as follows:
1)
THe authors used "A time series dataset of CCL5 expression, comprising 230 liver cancer patients". However this is not sufficient.
The dataset is relatively small for training deep learning models.
Although cross-validation and loss curves are provided, there is still a concern about overfitting.
2)
There is no comparative work with the literature or other similar algorithms.
More meaningful comparisons would be with other machine learning approaches
3)
To evaluate the accuracy of forecasting models,why don't you use Mean Absolute Error (MAE), Mean Absolute Percentage Error (MAPE), and Root Mean Square Error (RMSE) in the system?
4) Figures are not constructed in an appropriate format for an academic paper.
Figure captions are too long.
Additionally there are explanations in the figure captions.
Figure 1 is not clear.
in figure 1 .A what is the unit of Time?
What do you mean with the CCL% expression?
Figure 4 is not clear.
It should be redrawn
5)
THis is a research article. And it focus on the "A Machine Learning Framework for Cancer Prognostics". However there is no framework design in the paper!
How do you design the proposed system?
What is your parameteres here?
MAJOR COMMENT 1)
The manuscript cites only eight references, which is insufficient for an academic paper and does not meet the expected scholarly standards.
MAJOR COMMENT 2)
In the paper the authors said that "Throughout the middle portion of the time series, the fitted values (red line) continue to track the original data (blue line) with reasonable accuracy (Fig. 1A)."
What do you mean with the reasonable accuracy?
What is your metric for this?
In an academic approach it must be measurable and shown in the paper.
MAJOR COMMENT 3)
Equation are mentioned as text in the paper.
Δyt = c+Ï•1Δyt−1+Ï•2Δyt−2+Ï•3Δyt−3+Ï•4Δyt−4+Ï•5Δyt−5+ϵt,
The authors should give an equation number and use them with their references.
Reviewer 2 Report
Comments and Suggestions for Authors
Check the attached file.
